# SoftEDA: Rethinking Rule-Based Data Augmentation with Soft Labels

**Juhwan Choi, Kyohoon Jin, Junho Lee, Sangmin Song, Youngbin Kim**
Chung-Ang University, Seoul, Republic of Korea
`{gold5230,fhzh123,jhjo32,s2022120859,ybkim85}@cau.ac.kr`

## Abstract

Rule-based text data augmentation is widely used for NLP tasks due to its simplicity. However, this method can potentially damage the original meaning of the text, ultimately hurting the performance of the model. To overcome this limitation, we propose a straightforward technique for applying soft labels to augmented data. We conducted experiments across seven different classification tasks and empirically demonstrated the effectiveness of our proposed approach. We have publicly opened our source code for reproducibility.

## 1 Introduction

Data augmentation is a common strategy for mitigating overfitting and enhancing the robustness and generalizability of a model by augmenting the existing data or generating new data for training. Rule-based data augmentation is a prevalent approach to data augmentation that involves modifying the current data according to predefined policies. Examples of such policies for image data include rotation, flipping, and cropping (Yang et al., 2022).

Rule-based data augmentation methods are also frequently utilized for text data due to their simplicity and efficiency. One such method is easy data augmentation (EDA) (Wei & Zou, 2019), which consists of four operations: synonym replacement, random insertion, random swap, and random deletion. These operations enable the generation of augmented data that are similar to the original data, but with small variations. However, some researchers have raised concerns that the EDA method may hurt the meaning of a sentence. As an alternative, they proposed "an easier data augmentation" (AEDA) (Karimi et al., 2021), which is based solely on the random insertion of punctuation marks in {".", ";", "?", ":", "!", ","}.

In this work, we aimed to improve existing rule-based text data augmentation methods and propose a novel, yet simple data augmentation strategy called easy data augmentation with soft labels (softEDA). Traditional EDA techniques generate noisy and perturbed data from the original text by randomly inserting, swapping, or deleting words. This is different from rule-based image data augmentation methods, such as cropping and rotating, which preserve the fundamental semantics and thus allow original labels for augmented images to be maintained. However, in traditional EDA, these perturbed data keep the original one-hot label, making it *coarse* to the model and potentially reducing performance as a result.

To mitigate this performance degradation, softEDA incorporates noise-to-label values of the noisy augmented data. Label smoothing (Szegedy et al., 2016) is applied to the original data labels to acquire noisy labels. We assign these soft labels to augmented data and organize them with original data for training. By leveraging this straightforward technique, the model can enhance its robustness and performance by learning the relatively weaker signal of a soft label instead of the one-hot label.

To the best of our knowledge, this is the first study to introduce soft labels into rule-based text data augmentation methods. We evaluated softEDA on seven different text classification tasks and empirically showed its effectiveness compared to the previously proposed EDA and AEDA. We have opened our source code for further study and reproducibility.[1]

---

[1] `https://github.com/c-juhwan/SoftEDA`

Table 1: Accuracy (%) and gain (%p) across seven datasets. For softEDA, we denoted the best outputs among various smoothing values. Full results with each $\alpha$ can be found in Appendix C.

| | **Dataset** | | | | | | |
| Model | SST2 | CR | MR | TREC | SUBJ | PC | CoLA |
|---|---|---|---|---|---|---|---|
| CNN w/o Aug | 77.84 | 77.29 | 74.94 | 86.76 | 90.18 | 91.62 | 69.13 |
| w/ EDA | +0.12 | -0.02 | +0.70 | +0.15 | +0.05 | +0.81 | -1.50 |
| w/ AEDA | +0.72 | -0.36 | -0.40 | +0.90 | -0.50 | +0.04 | -2.23 |
| w/ softEDA | **+0.83** | **+0.91** | **+1.84** | **+2.03** | **+0.99** | **+1.29** | **+0.21** |
| LSTM w/o Aug | 75.80 | 74.26 | 74.05 | 86.37 | 88.84 | 92.74 | 69.18 |
| w/ EDA | +0.82 | +1.70 | +0.70 | -3.24 | +1.69 | +0.49 | -0.07 |
| w/ AEDA | **+2.97** | +0.28 | +0.49 | -1.37 | +0.64 | -0.15 | -0.37 |
| w/ softEDA | +2.59 | **+2.90** | **+1.41** | **+1.95** | **+2.18** | **+0.60** | **+0.18** |
| BERT w/o Aug | 89.74 | 89.08 | 84.28 | 95.47 | 96.18 | 93.44 | 75.38 |
| w/ EDA | +0.71 | -0.41 | -0.92 | +0.51 | -0.35 | +0.58 | -0.45 |
| w/ AEDA | +0.22 | +1.84 | **+0.19** | -0.67 | -0.30 | -0.15 | -0.34 |
| w/ softEDA | **+0.83** | **+2.10** | **+0.19** | **+1.17** | **+0.15** | **+0.67** | **+1.50** |

## 2 METHOD

As EDA generates data through perturbing original data, augmented data may have uncertainty compared to original data. Despite this uncertainty, EDA assigns the exact same one-hot label to augmented data. SoftEDA, on the other hand, adheres to the fundamental concept of EDA and employs its four sub-operations. However, unlike EDA, softEDA introduces uniform noise distribution across every class through label smoothing considering the uncertainty. For data $(\boldsymbol{x}, \boldsymbol{y})$ in dataset $D$, the label of augmented data $\hat{\boldsymbol{y}}$ is defined according to the following equation:

$$\hat{\boldsymbol{y}} = (1 - \alpha)\boldsymbol{y} + \frac{\alpha}{N_{Class}} = \begin{cases} (1 - \alpha) + \frac{\alpha}{N_{Class}} & if \ y = y_i \\ \frac{\alpha}{N_{Class}} & Otherwise \end{cases}$$

## 3 EXPERIMENT

We evaluated softEDA with seven different text classification tasks. To assess the effectiveness of softEDA, we constructed text classification models based on convolutional neural networks (CNN) (Kim, 2014), long short-term memory (LSTM) (Liu et al., 2016), and pretrained BERT (Devlin et al., 2019). For our experiments, we used $\alpha = [0.1, 0.15, 0.2, 0.25, 0.3]$ to determine the optimal value for each model. Further details about experimental setup can be found in Appendix A and B.

The results of the experiment are provided in table 1. We found that, in most cases, softEDA significantly improved the performance of the model compared to other methods. Furthermore, softEDA was able to boost the model even when other methods exhibited degrading performance.

It is noteworthy that softEDA demonstrated promising performance on the Corpus of Linguistic Acceptability (CoLA) (Warstadt et al., 2019) dataset, which measures the grammatical acceptability of given sentences. While EDA and AEDA failed to achieve good result because they use one-hot labels and do not maintain the syntactic structure of a sentence, softEDA improved performance by manipulating the label of augmented data.

## 4 CONCLUSION

We have shown that rule-based text data augmentation methods can be refined by applying soft labels to augmented data. By using this simple approach, we found it possible to further boost existing rule-based data augmentation methods with a simple modification. Future work could extend the softEDA approach to other rule-based methods, including AEDA and character-level noise injection (Belinkov & Bisk, 2018), and could include an investigation of the best parameters for each method.

ACKNOWLEDGEMENTS

We would like to express our sincere gratitude to the reviewers and organizers of this track. This research was supported by Basic Science Research Program through the National Research Foundation of Korea(NRF) funded by the Ministry of Education(NRF-2022R1C1C1008534), and Institute for Information & communications Technology Planning & Evaluation (IITP) through the Korea government (MSIT) under Grant No. 2021-0-01341 (Artificial Intelligence Graduate School Program, Chung-Ang University).

URM STATEMENT

First author Juhwan Choi meets the URM criteria of ICLR 2023 Tiny Papers Track. He is outside the range of 30-50 years, non-white researcher. This work is his first submission and accepted paper at ICLR or other top-tier conferences.

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

## A  DATASET SPECIFICATIONS

Table 2: Dataset used for the experiment.

| Dataset | Dataset | | | |
| --- | --- | --- | --- | --- |
| | Task | $N_{Class}$ | $N_{Train}$ | $N_{Test}$ |
| SST2 (Socher et al., 2013) | Sentiment | 2 | 6,919 | 1,820 |
| CR (Hu & Liu, 2004) (Liu et al., 2015) | Sentiment | 2 | 3,011 | 752 |
| MR (Pang et al., 2002) | Sentiment | 2 | 9,593 | 1,067 |
| TREC (Li & Roth, 2002) | Question Type | 6 | 5,452 | 500 |
| SUBJ (Pang & Lee, 2004) | Subjectivity | 2 | 8,000 | 2,000 |
| PC (Ganapathibhotla & Liu, 2008) | Pro-Con | 2 | 39,418 | 4,506 |
| CoLA (Warstadt et al., 2019) | Linguistic Acceptability | 2 | 8,551 | 527 |

## B  EXPERIMENTAL SETUP

In this section, we demonstrate experimental setups and implementation details for reproduction. We have opened our source code for further information.

**SoftEDA Implementation.** SoftEDA leverages all four sub-operations of EDA for augmentation and involves three distinct steps. Firstly, a sub-operation is randomly selected from the available four operations. Subsequently, the chosen operation is applied to a given sentence to attain augmented sentence. Finally, in contrast to conventional EDA techniques, SoftEDA applies label smoothing to the label of augmented data to obtain a soft label. It is recommended to refer to the accompanying code for further information.

**Models.** For the LSTM model, we used bidirectional two-layer LSTM and passed extracted hidden states to the linear classifier. For the CNN model, we mainly followed the core architecture of Kim (2014) to extract features. For the BERT model, we used the bert-base-uncased model at Hugging Face[2]. Extracted features from the models were passed to the linear classifier, with a hidden size of

---

[2] https://huggingface.co/bert-base-uncased

768 and dropout with $p = 0.2$, followed by a gaussian error linear unit (Hendrycks & Gimpel, 2016) activation and the final linear layer.

**Training Hyperparameters.** We used AdamW (Loshchilov & Hutter, 2019) as the optimizer, with a learning rate of 1e-4 and weight decay of 1e-5. There was no scheduler used. We trained every model with a batch size of 32 and applied early stopping with a patience of 5 epochs.

**Other Details.** We randomly selected 20% of the training data as a validation set if there was no predefined validation set. Every model used the BERT tokenizer from Hugging Face, with a maximum token length of 100. The training was conducted using a single NVIDIA RTX 3090 GPU.

## C    FULL EXPERIMENT RESULTS

Table 3: Accuracy (%) of the full experiment results.

| Model | SST2 | CR | MR | TREC | SUBJ | PC | CoLA |
|---|---|---|---|---|---|---|---|
| CNN w/o Augmentation | 77.84 | 77.29 | 74.94 | 86.76 | 90.18 | 91.62 | 69.13 |
| w/ EDA | 77.96 | 77.27 | 75.64 | 86.91 | 90.23 | 92.43 | 67.63 |
| w/ AEDA | 78.56 | 76.93 | 74.54 | 87.66 | 89.68 | 91.66 | 66.90 |
| w/ softEDA $\alpha = 0.1$ | **78.67** | 76.13 | 74.88 | 87.50 | 90.33 | 92.61 | 68.05 |
| w/ softEDA $\alpha = 0.15$ | 77.07 | 75.48 | 73.99 | **88.79** | 89.88 | **92.91** | 68.79 |
| w/ softEDA $\alpha = 0.2$ | 77.90 | **78.20** | 74.75 | 87.73 | 90.13 | 92.35 | 68.79 |
| w/ softEDA $\alpha = 0.25$ | 78.10 | 77.91 | 75.18 | 88.01 | 90.67 | 92.49 | 68.97 |
| w/ softEDA $\alpha = 0.3$ | 75.48 | 76.10 | **76.78** | 87.54 | **91.17** | 92.69 | **69.34** |
| LSTM w/o Augmentation | 75.80 | 74.26 | 74.05 | 86.37 | 88.84 | 92.74 | 69.18 |
| w/ EDA | 76.62 | 75.96 | 74.75 | 83.13 | 90.53 | 93.23 | 69.11 |
| w/ AEDA | **78.77** | 74.54 | 74.54 | 85.00 | 89.48 | 92.59 | 68.81 |
| w/ softEDA $\alpha = 0.1$ | 78.39 | **77.16** | 73.62 | 87.46 | 88.94 | 93.12 | 69.18 |
| w/ softEDA $\alpha = 0.15$ | 74.15 | 72.99 | 73.71 | 86.68 | 89.93 | **93.34** | 69.18 |
| w/ softEDA $\alpha = 0.2$ | 76.09 | 75.73 | 73.01 | **88.32** | 89.53 | 93.07 | 69.00 |
| w/ softEDA $\alpha = 0.25$ | 77.23 | 75.48 | 71.48 | 86.84 | 90.48 | 92.92 | 67.34 |
| w/ softEDA $\alpha = 0.3$ | 77.18 | 75.61 | **75.46** | 86.09 | **91.02** | 92.63 | **69.36** |
| BERT w/o Augmentation | 89.74 | 89.08 | 84.28 | 95.47 | 96.18 | 93.44 | 75.38 |
| w/ EDA | 90.45 | 88.67 | 83.36 | 95.98 | 95.83 | 94.02 | 74.93 |
| w/ AEDA | 89.96 | 90.92 | **84.47** | 94.80 | 95.88 | 93.29 | 75.04 |
| w/ softEDA $\alpha = 0.1$ | 89.63 | 89.37 | 83.18 | 94.02 | **96.33** | 93.87 | 76.72 |
| w/ softEDA $\alpha = 0.15$ | 89.52 | 89.74 | 83.82 | 95.00 | 95.68 | 93.43 | 75.40 |
| w/ softEDA $\alpha = 0.2$ | 89.62 | **91.18** | **84.47** | 95.20 | 96.23 | 93.87 | 76.19 |
| w/ softEDA $\alpha = 0.25$ | 89.51 | **91.18** | 83.36 | **96.64** | 96.08 | **94.11** | **76.88** |
| w/ softEDA $\alpha = 0.3$ | **90.57** | 88.18 | 82.48 | 94.69 | 96.18 | **94.11** | 75.61 |

Table 4: F1 scores of the full experiment results.

| **Model** | SST2 | CR | MR | TREC | SUBJ | PC | CoLA |
|---|---|---|---|---|---|---|---|
| CNN w/o Augmentation | .7726 | .7351 | .4350 | .8400 | .8982 | .5165 | .4516 |
| w/ EDA | .7749 | .7532 | **.4520** | .8278 | .8982 | .5328 | .5262 |
| w/ AEDA | .7799 | .7325 | .4489 | .8593 | .8933 | .5273 | .4821 |
| w/ softEDA $\alpha = 0.1$ | **.7805** | .7418 | .4334 | .8430 | .8993 | .5369 | .4595 |
| w/ softEDA $\alpha = 0.15$ | .7645 | .7287 | .4481 | **.8726** | .8951 | **.5449** | **.5500** |
| w/ softEDA $\alpha = 0.2$ | .7740 | **.7558** | .4366 | .8516 | .8977 | .5397 | .4240 |
| w/ softEDA $\alpha = 0.25$ | .7747 | .7551 | .4379 | .8489 | .9037 | .5399 | .4206 |
| w/ softEDA $\alpha = 0.3$ | .7481 | .7266 | .4434 | .8422 | **.9076** | .5440 | .4407 |
| LSTM w/o Augmentation | .7528 | .6978 | .4369 | .8204 | .8839 | .5373 | .4070 |
| w/ EDA | .7592 | .7219 | .4365 | .7645 | **.9105** | .5420 | **.4847** |
| w/ AEDA | **.7827** | .7173 | .4346 | .8230 | .8908 | .5263 | .4230 |
| w/ softEDA $\alpha = 0.1$ | .7787 | **.7548** | .4446 | .8521 | .8856 | **.5558** | .4070 |
| w/ softEDA $\alpha = 0.15$ | .7354 | .6932 | .4465 | **.8564** | .8954 | .5492 | .4070 |
| w/ softEDA $\alpha = 0.2$ | .7563 | .7394 | .4323 | .8525 | .8922 | .5521 | .4190 |
| w/ softEDA $\alpha = 0.25$ | .7564 | .7239 | .4229 | .8403 | .9010 | .5517 | .4193 |
| w/ softEDA $\alpha = 0.3$ | .7670 | .7335 | **.4517** | .8412 | .9075 | .5439 | .4139 |
| BERT w/o Augmentation | .8942 | .8820 | .4812 | .9475 | .9606 | .5605 | .6779 |
| w/ EDA | .9012 | .8775 | .4798 | .9430 | .9569 | .5738 | .6749 |
| w/ AEDA | .8963 | .9017 | .4839 | .9340 | .9573 | .5495 | .6802 |
| w/ softEDA $\alpha = 0.1$ | .8932 | .8849 | **.4925** | .9274 | **.9617** | .5543 | **.6998** |
| w/ softEDA $\alpha = 0.15$ | .8910 | .8890 | .4820 | .9345 | .9550 | .5569 | .6883 |
| w/ softEDA $\alpha = 0.2$ | .8928 | **.9036** | .4830 | .9329 | .9610 | **.5831** | .6934 |
| w/ softEDA $\alpha = 0.25$ | .8916 | .9030 | .4797 | **.9655** | .9596 | .5656 | .6985 |
| w/ softEDA $\alpha = 0.3$ | **.9023** | .8691 | .4614 | .9341 | .9601 | .5586 | .6659 |

