# OpenReview forum: "SoftEDA: Rethinking Rule-Based Data Augmentation with Soft Labels"
_ICLR.cc/2023/TinyPapers — Submitted to Tiny Papers @ ICLR 2023_

### Official Review · Reviewer_ySa5 · 2023-03-20

**Confidence:** 3

**Summary Of Contributions:**

This work proposes SoftEDA, which follows the concept of easy data augmentation (EDA) to augment text data by perturbation then improves upon it to handle the uncertainty in the generated data augmentation. SoftEDA introduces uniform noise distribution across every class through label smoothing to provide the supervision in the form of soft labels instead of one-hot labels.

**Rating:**

High Potential (HP): a submission which meets the reviewing criteria and has potential to make an impact on the field

**Strengths And Weaknesses:**

Strengths
- The paper is well-written and easy to read.
- The paper clearly presents a problem, explains the solution, and provides empirical results of the proposed solution on various datasets and model architectures.
- Experimental settings are described in detail. The code will also be made publicly available to encourage reproducibility.

Weaknesses
- Although the paper identifies the previous works’ (i.e., EDA and AEDA) inability to maintain sentence’s syntactic structure as a weakness, it is not clear how SoftEDA introduces any difference when it comes to this.

**Suggested Changes:**

- While it is understandable that Table 1 is a summarized overview of the results, Table 1 still should denote the 𝛼 used by SoftEDA for clarity.
- In the case of SoftEDA also being incapable of maintaining syntactic structure, it might be better to highlight SoftEDA’s advantages using another of the previous works’ existing gaps. In the case of SoftEDA being capable, it would be interesting to see further analysis that shows how the supervision of soft labels could be more beneficial compared to the hard labels for maintaining the syntactic structure.

---

### Official Review · Reviewer_1Gct · 2023-03-29

**Confidence:** 4

**Summary Of Contributions:**

This paper presents softEDA, a simple and novel method to augment text data and improve the performance of trained models. Advancing prior work, softEDA adds label smoothing to augmented samples. Experiment results highlight that softEDA significantly improves the model's performance comparing to existing augmentation methods.

**Rating:**

High Impact (HI): a submission which meets the reviewing criteria and is predicted to make an impact on the field

**Strengths And Weaknesses:**

## Strengths and weaknesses

#### Clarity

1. Overall the paper is clear and well-written. I enjoy reading it!

#### Correctness

1. The claims and conclusions are justified by the findings.

#### Reproducibility

1. The paper includes findings from an empirical experiment. The code is open-sourced.

#### Follows basic requirements

1. This paper follows the basic formatting requirements.

**Suggested Changes:**

## Suggested changes

1. In the experiments, it is unclear what traditional EDA methods are used before applying smooth labeling for softEDA. In other words, does softEDA use EDA or AEDA, or both?
2. SoftEDA is a simple method that can be generalized to a wide range of augmentation techniques. For future work, I would be very interested to see how it improves other types of augmentation methods (e.g., NL Augmenter[1]).

[1] Dhole, Kaustubh D., et al. "Nl-augmenter: A framework for task-sensitive natural language augmentation." arXiv preprint arXiv:2112.02721 (2021).

---

### Author Response · Authors · 2023-05-30
**Reply to reviewers**

We would like to extend our sincere gratitude to the reviewers for their valuable feedback on our paper.

We have carefully considered the suggestions provided by the reviewers and have made the revisions accordingly:

1. Detailed information about the process of softEDA: We included an explanation of the softEDA methodology in Appendix A, providing a better understanding of its implementation and process.

2. Update Table 1 for clarity: We have made a minor adjustment to Table 1 to improve clarity. We have mentioned in the paper that readers can refer to Appendix C for a more detailed analysis of the results with each 𝛼 value.

Thank you once again for your time and consideration of the paper.

---

### Comment · Area_Chair_mKaN · 2023-06-01
**This work meets the threshold for archival, contains the URM statement and is deanonymized**

---

### Meta-Review · Area_Chair_mKaN · 2023-04-08

**Recommendation:** Invite to present (notable)
**Confidence:** 5

**Metareview:**

The authors proposed combining two established methods: easy data augmentation (EDA) and label smoothing. In a series of experiments, they find that this combined method outperforms existing methods across multiple benchmarks. As all the reviewers mentioned, this paper is clear, correct, and reproducible. The reviewers shared minor suggested changes regarding clarity, but these should be easy to address and overall do not impact the quality or potential impact of this work (see additional comments below).

**Summary:**

The authors propose using label smoothing in conjunction with language data augmentation and demonstrate improvements over other methods across multiple benchmarks. Reviewers agree this is a clear, correct, and reproducible submission with only minor suggested changes.

**Comments And Feedback To The Authors:**

Space-permitting, please consider making the clarity changes suggested by the reviewers:
- details of which EDA method SoftEDA builds on (I found this in the supplied code but it is not clear from the paper itself, nor the appendix)
- potential updates to table 1

**Reason For Not Giving A Higher Recommendation:**

N/A

**Reason For Not Giving A Lower Recommendation:**

All reviewers agreed this submission is clear, correct, and reproducible with only minor changes suggested

---

### Decision · Program_Chairs · 2023-04-09

Invite to present (notable)